# Effect of the COVID-19 Pandemic on Suicide Mortality in Brazil: An Interrupted Time Series Analysis

**DOI:** 10.3390/ijerph22020138

**Published:** 2025-01-21

**Authors:** Karina Cardoso Meira, Raphael Mendonça Guimarães, Rafael Tavares Jomar, Cosme Marcelo Furtado Passos da Silva, Fabiana Serpa Braiti, Eder Samuel Oliveira Dantas

**Affiliations:** 1Department of Pharmaceutical Sciences, Federal University of São Paulo, Diadema 09913-030, SP, Brazil; 2Oswaldo Cruz Foundation, National School of Public Health, Rio de Janeiro 21041-210, RJ, Brazil; raphael.guimaraes@fiocruz.br (R.M.G.); cosme.passos@fiocruz.br (C.M.F.P.d.S.); 3Assistance Coordination, National Cancer Institute, Rio de Janeiro 20230-130, RJ, Brazil; rafaeljomar@yahoo.com.br; 4University City Campus, Paulista University, São Paulo 05510-030, SP, Brazil; fabianauti@hotmail.com; 5Onofre Lopes University Hospital, Federal University of Rio Grande do Norte, Natal 59012-300, RN, Brazil; edersamuel_rn@hotmail.com

**Keywords:** suicide, health inequities, mental health, health crisis

## Abstract

This study analyzed the effect of the COVID-19 pandemic on suicide rates among Brazilian residents, stratified by sex. It examined monthly suicide rates using interrupted time series analysis. Researchers compared the months before the pandemic (January 2017 to February 2020) with those after the first diagnosed case of COVID-19 in Brazil (March 2020 to December 2023). They applied an interrupted time series model (quasi-Poisson) to account for serial autocorrelation in the residuals and seasonality. During this period, authorities reported 102,081 suicides in Brazil. The age-standardized annual suicide rate among men was 3.71 times higher than the rate among women (12.33 suicides per 100,000 vs. 3.32 suicides per 100,000 women). The South and Midwest regions had the highest standardized annual average suicide rates. Suicide rates among men dropped abruptly at the pandemic’s onset (RR < 1, *p* < 0.05). However, Black men, women aged 15 to 19 years, and elderly individuals exhibited a significant increase (RR > 1, *p* < 0.05). Over time, suicide rates rose across most age groups, regions, and methods studied (RR > 1, *p* < 0.05). The pandemic’s impact differed significantly between men and women based on stratification variables. Nonetheless, a progressive upward trend emerged throughout the pandemic.

## 1. Introduction

Suicide represents a complex and multifaceted social issue and poses a significant global public health challenge. Each year, approximately 700,000 lives are lost to suicide worldwide. Brazil ranks among the top ten countries with the highest absolute number of suicides, reporting 15,507 cases in 2021. The country’s suicide rate is 7.5 deaths per 100,000 inhabitants [1]. Although this rate is lower than those observed in many Global North countries, such as those in Europe, North America, and Asia, where rates often exceed 15 per 100,000, Brazil surpasses countries in South America like Argentina (8.4/100,000), Colombia (3.9/100,000), and Peru (2.8/100,000). However, smaller nations like Guyana (40.3/100,000) and Suriname (25.4/100,000) exhibit significantly higher rates [2].

While suicide rates have declined in many regions globally, they have risen in Brazil and the Americas in recent years [1,2]. Within Brazil, regional disparities are pronounced. For example, in 2021, the South region reported 11.0 deaths per 100,000 inhabitants, whereas the Southeast region reported 6.34 per 100,000 [2]. Variations in cultural factors, lifestyle differences, and disparities in health information quality contribute to these differences.

Suicidal behavior is linked to a combination of individual and situational risk factors. At the individual level, mental disorders such as depression, bipolar disorder, schizophrenia, and substance abuse are prominent. Situational factors, including socio-economic inequalities, limited access to healthcare, economic crises, and health emergencies, also play significant roles. Moreover, the availability of lethal means, such as firearms and toxic pesticides, increases the risk of suicide [3,4,5,6,7,8,9,10].

The emergence of the COVID-19 pandemic led some researchers to predict an increase in self-harm, suicide attempts, and completed suicides during the pandemic [8,10,11,12]. Preventive measures such as lockdowns, school closures, and healthcare service reconfigurations, while effective in reducing disease transmission, exacerbated psychosocial stressors and economic instability [13]. This worsened pre-existing mental health issues, particularly in vulnerable populations [14,15,16,17,18].

The pandemic’s impact on suicide rates was uneven across countries, influenced by socio-economic and cultural characteristics, as well as individual factors. High-income countries that quickly implemented financial support measures for their populations, such as Germany, Australia, Canada, England, South Korea, Japan, Norway, and Taiwan, often observed reductions or stabilization in suicide rates [19,20,21,22,23]. Investments in social welfare, healthcare, and employment programs in these nations helped mitigate the mental health impacts of economic crises. For example, in Europe and USA, each additional USD 10 per capita invested in employment programs during economic downturns reduced unemployment’s impact on suicide rates [19,20,21,22,23].

Conversely, in countries such as Nepal and Brazil, suicide rates increased during the initial months of the pandemic [24,25]. Unemployment and poverty, which tend to worsen during such periods, appear to have significantly contributed to this rise. Evidence also shows that after public health crises, natural disasters, terrorist attacks, and wars, suicide rates often increase over time. This pattern is particularly evident in low- and middle-income countries, where public support policies are frequently delayed in their implementation [11,22,26,27,28,29,30].

In Brazil, the government’s income redistribution policy (emergency aid) was implemented late. Moreover, accessing this financial assistance required individuals to register through a mobile phone application. This requirement posed a significant barrier for populations lacking internet access. Compounding these challenges, during the second year of the pandemic, the government reduced the value of this financial aid and eventually discontinued it altogether. These factors likely exacerbated economic hardships and contributed to the observed rise in suicide rates [24,25,30,31,32,33,34,35,36,37,38].

The pandemic’s impact on suicide rates varied by sex, age, ethnicity, and geography [19,20,21,22,23,24,25,30,31,32,33,34,35]. In Brazil, despite a high gross domestic product and universal healthcare, socio-economic and health inequities remain significant. These inequities deepened during the pandemic due to the denialist stance of the previous government, leading to high COVID-19 mortality, unemployment, food insecurity, and worsening mental health conditions such as stress, anxiety, and depression [36,37,38].

Studies analyzing suicide records in Brazil from 2020 to early 2021 have yielded conflicting results. While Orellana and De Souza (2022) [34] and Soares et al. (2022) [39] reported reductions in suicide during the pandemic’s early months, Ornell et al. (2022) [26] found an increase in suicide-related deaths, particularly in the Midwest, Northeast, and Southeast regions. Orellana and De Souza (2022) later identified demographic-specific increases, including a 26% rise among men aged 60 and older in the North region and a 40% rise among women of the same age group in the Northeast region [34]. These discrepancies may reflect variations in study methodologies and the limited timeframes analyzed [26,34,39].

The initial reduction in suicides documented after such emergencies is attributed to a temporary increase in social cohesion and solidarity within affected societies. This surge in communal bonds serves as a protective factor against suicide. However, over time, this heightened sense of social cohesion tends to erode. As these bonds weaken, suicide rates subsequently increase [11,26,27,28,29].

This study seeks to deepen the understanding of the pandemic’s impact on suicide rates in Brazil. Specifically, it analyzes data stratified by age group, geographic region, method of suicide, and race/skin color, disaggregated by sex. The analysis spans the entirety of the pandemic period, from March 2020 to December 2022, as well as the post-health-emergency phase, from May 2022 to December 2023.

The central research question focuses on whether the COVID-19 pandemic differentially affected suicide rates among men and women in Brazil based on age group, region, race/skin color, and method of suicide. Addressing this question requires a nuanced understanding of societal gender roles, as these roles significantly influence self-destructive behaviors. These behaviors are shaped by hegemonic masculinities and femininities, which contribute to the “suicide or gender paradox” [40,41,42]. Men often resort to more lethal methods, such as hanging or firearms, resulting in higher mortality rates. In contrast, women display higher rates of suicidal ideation and non-lethal attempts, frequently employing methods like self-intoxication [40,41,42,43,44,45,46,47,48].

The overarching aim of this research is to analyze the COVID-19 pandemic’s impact on suicide rates among Brazilian residents. This analysis is stratified by sex, region, age group, race/skin color, and method, covering the period from January 2017 to December 2023. An interrupted time series analysis is employed to capture the temporal shifts in suicide rates, providing a comprehensive view of the pandemic’s effects on this critical public health issue.

## 2. Materials and Methods

### 2.1. Study Design and Location

This interrupted time series (ITS) ecological study followed the Guidelines for Accurate and Transparent Health Estimates Reporting (GATHER) statement [49]. It aimed to evaluate the behavior of monthly suicide rates in Brazil over 84 months. Of these, 38 months preceded the first confirmed COVID-19 case (January 2017 to February 2020), while 46 months followed the initial diagnosis (March 2020 to December 2023). The first confirmed case of COVID-19 in Brazil was reported on 26 February 2020.

Brazil is divided into five major geographical regions: North, Northeast, Southeast, South, and Midwest (Figure 1), with an estimated population of 207.8 million inhabitants in 2022. The South region comprises the states of Paraná (PR), Santa Catarina (SC), and Rio Grande do Sul (RS). The Southeast region includes Espírito Santo (ES), Minas Gerais (MG), Rio de Janeiro (RJ), and São Paulo (SP). The North region consists of Acre (AC), Amazonas (AM), Amapá (AP), Pará (PA), Rondônia (RO), Roraima (RR), and Tocantins (TO). The Northeast region encompasses Alagoas (AL), Bahia (BA), Ceará (CE), Maranhão (MA), Paraíba (PB), Pernambuco (PE), Piauí (PI), and Sergipe (SE). The Midwest region includes the Federal District (Brasília) and three states: Goiás (GO), Mato Grosso (MT), and Mato Grosso do Sul (MS) (Figure 1).

The Southeast region has the largest population, followed by the Northeast. In contrast, the North and Midwest regions have the smallest populations. The highest Human Development Index (HDI) values are observed in the South region, while the lowest values are concentrated in the Northeast and North regions. The North region is marked by low population density and a vast geographic area, including much of the Amazon rainforest. The Southeast region excels due to its dynamic labor market. Meanwhile, the Midwest region, home to the nation’s capital, primarily relies on agriculture and livestock farming [50].

### 2.2. Data Source

We extracted suicide data from the Mortality Information System (SIM), utilizing records of deaths from external causes provided by the Informatics Department of the Brazilian Unified Health System (SUS) [51]. According to the 10th International Statistical Classification of Diseases and Related Health Problems (ICD-10), the selected codes were X60–X84, representing intentional self-inflicted injuries. We adhered to the Ministry of Health’s methodology, which defines suicide as the sum of deaths classified under X60–X84 and Y87.0 (sequelae of intentional self-inflicted injury) [2].

We obtained population data from the SUS Informatics System (DataSUS) based on national censuses from 1980, 1991, 2000, 2010, and 2022. For the intervening years, we utilized intercensal projections for July 1st, estimated by IBGE [50]. Additionally, we incorporated population estimates by race/skin color from the Economic Research Institute (IPEA), particularly from their publication “The Portrait of Gender and Race Inequalities”. However, race/skin color data disaggregated by sex were available only for three categories: White, Black, and Brown [52].

Research consistently demonstrates that suicidal behavior exhibits significant differences between men and women [11,22,32,40,41,42,45,46,47,48]. These differences are often exacerbated during periods of crisis. Consequently, we stratified all analyses by sex. We also included variables such as age group (10–14, 15–19, 20–39, 40–59, and 60 years or older), macro-region (North, Northeast, Midwest, Southeast, and South), and race/skin color (White and Black). Regarding suicide methods, we focused on the most prevalent approaches in Brazil: self-inflicted poisoning (X60–X69), hanging, strangulation, and suffocation (X70), and firearm use (X72–X74) [2].

The decision to commence the analysis with the 10–14 age group was deliberate. Suicides are exceedingly rare in younger age groups, and deaths among children under 10 years old may involve situations where they did not fully comprehend the irreversible nature of death [2,25,31,32,33,34].

Finally, concerning race/skin color, we adhered to the Brazilian Institute of Geography and Statistics (IBGE) classification. The Black population encompasses individuals who self-identify as Black or Brown [52], aligning with Brazil’s categorization of African descendants.

### 2.3. Data Analysis

#### Exploratory and Bivariate Analysis

We calculated annual and monthly age-specific suicide rates by sex and standardized these rates using the direct method, adopting the world population proposed by the World Health Organization as the standard [53]. Afterward, we estimated standardized annual mean mortality rates for each year and the overall period (2017 to 2023), stratified by geographic region and sex.

For the other variables, we calculated standardized mean monthly rates, dividing the data into two periods: the 38 months preceding the first confirmed COVID-19 case (January 2017 to February 2020) and the 46 months following the initial diagnosis (March 2020 to December 2023). These rates were further stratified by age group, region, method of suicide, and race/skin color.

After estimating these rates, we calculated the mean monthly rates and standard deviations for the entire period, as well as for the months preceding and the months following the pandemic (March 2020 to December 2023).

To compare differences across age groups, regions, methods of suicide, and race/skin color, we used Welch’s test and ANOVA. These tests were deemed appropriate based on the results from the Shapiro–Wilk and Kolmogorov–Smirnov tests, which indicated a normal distribution. When ANOVA results were statistically significant, we conducted post hoc comparisons using Tukey’s test.

Tukey’s test, also known as Tukey’s Honestly Significant Difference (Tukey’s HSD), is a multiple comparison procedure applied after ANOVA. It identifies which group means differ significantly while controlling the overall (Type I) error rate during multiple comparisons.

The Welch test, a variation of the *t*-test adapted for unequal sample sizes, was particularly suitable given the nature of our dataset, which spanned 38 months before the first confirmed COVID-19 case and 46 months following the initial diagnosis [54].

To examine the temporal evolution of standardized monthly suicide rates by sex and stratified variables, we created line graphs and applied LOESS (Locally Estimated Scatterplot Smoothing) to smooth the trends [55]. LOESS is a nonparametric local polynomial regression method that assigns greater weights to nearby observations and smaller or zero weights to distant ones, usually controlled by a smoothing band. The model fits low-order polynomials locally to capture smooth variations without enforcing the rigidity of a global parametric function. Additionally, local fitting is typically performed using weighted least squares, with optional iterations to address outliers robustly.

### 2.4. An Analysis of the Impact of the Pandemic on Suicides in Brazil Using Interrupted Time Series

We analyzed the effect of the pandemic on suicides using an interrupted time series (ITS). Researchers widely employ this method in the literature to evaluate the impact of public policies and significant historical events on health outcomes [56,57,58,59,60,61,62,63]. This method involves algebraic segmentation of the time series, incorporating changes in both level and trend [56,62,63]. The level represents the initial value of the series in each segment. The trend indicates the change in values over the period covered by that segment. This approach identifies whether the intervention caused an immediate effect (level change) or a progressive impact (trend change) on the outcome.

We adjusted suicide mortality rates by a segmented regression model. The response variable was the number of suicides, while the independent variable included all months from 2017 to 2023 (ranging from 1 to 84 months). To examine level changes, we coded the pre-pandemic period as 0 and assigned the months following the pandemic a value of 1 [56,62,63]. We employed a quasi-Poisson regression model following three steps detailed in Table 1. This model adjusts the data to ensure the variance is proportional to the mean, correcting over-dispersion commonly found in count data. The level and trend change model (level and slope change) analyzed the abrupt change in the level and the gradual shift in the trend of monthly suicide rates after the onset of the COVID-19 pandemic. This analysis followed the Equation (1) [56,63].*Y_t_*∼Quasi-Poisson (*μ_t_*)(1)$$\log\left(\mu_{t}\right)=\beta_{0}+\beta_{1}T+\beta_{2}X_{t}+\beta_{3}\left(T-T_{0}\right)X_{t}+offset\left(\log\left(Popt\right)\right)$$
where *Y_t_* represents the observed count of suicides at time *t*, and *μ_t_* is the expected mean of suicides at time *t*. *T* represents the time in months since the beginning of the study period, while *X_t_* refers to the intervention (the pandemic), modeled as a dummy variable, where *t* = 0 for the pre-pandemic period and *t* = 1 for the period during the pandemic, beginning at the 39th month (March 2020). *β*_0_ represents the baseline level at *T* = 0. *β*_1_ captures the trend in suicide counts associated with the passage of time (the counterfactual trend). *β*_2_ relates to the change in the baseline level, i.e., the expected mean level of suicides post-pandemic, while *β*_3_ quantifies the change in slope following the onset of the pandemic (with *T*_0_ marking the pandemic onset). Lastly, log(Pop_t_) serves as an offset term that adjusts the expected mean of suicides based on population size [56,63].

In constructing the models, we examined the presence of seasonality. This analysis incorporated the month length and seasonal effects, modeled using Fourier terms. Specifically, we employed two pairs of sine and cosine functions with a 12-month periodicity, represented by the following Equation (2) [55,56,60,63].(2)$$\log\left(\mu_{t}\right)=\beta_{0}+\beta_{1}T+\beta_{2}X_{t}+\beta_{3}\left(T-T_{0}\right)X_{t}+offset\left(\log\left(Popt\right)\right)+\;\gamma_{1}sin\frac{2\pi X\left(i\right)}{L}+\gamma_{2}cos\frac{2\pi X\left(i\right)}{L}+\gamma_{3}sin\frac{4\pi X\left(i\right)}{L}+\gamma_{4}cos\frac{4\pi X\left(i\right)}{L}$$
where the Fourier terms are $\gamma_{1}sin\frac{2\pi X\left(i\right)}{L}$, $\gamma_{2}cos$ $\frac{2\pi X\left(i\right)}{L}$, $\gamma_{3}sin$ $\frac{4\pi X\left(i\right)}{L}$, and $\gamma_{4}cos$ $\frac{4\pi X\left(i\right)}{L}$. *L* represents the seasonality period (12 months); $\gamma_{1}e\gamma_{2}$ are coefficients associated with the fundamental frequency $\frac{2\pi X\left(i\right)}{L}$. $\gamma_{1}$ is the amplitude of the sinusoidal component of the fundamental frequency, $\gamma_{2}$ is the amplitude of the cosine component of the fundamental frequency, $\gamma_{3}e\gamma_{4}$ are coefficients associated with the first harmonic $\frac{4\pi X\left(i\right)}{L}{,\gamma }_{3}$ is the amplitude of the sinusoidal component of the first harmonic, and $\gamma_{4}$ is the amplitude of the cosine component of the first harmonic.

In the modeling process, we also analyzed the presence of autocorrelation in the residuals using the Durbin–Watson test and its critical values, along with the sample autocorrelation function and partial autocorrelation (ACF and PACF) plots [55,56,60,63,64]. If there is evidence of serial autocorrelation, one or more autoregressive terms can be added to the model, as indicated by significant peaks in the PACF, resulting in an AR (autoregressive) model [55,56,60,63,64].

The Durbin-Watson test produces a statistic typically ranging from 0 to 4, where values close to 2 indicate no autocorrelation, values near 0 suggest positive autocorrelation, and values near 4 indicate negative autocorrelation. A *p*-value can be associated with this test, indicating the presence of autocorrelation when *p* < 0.05. The Durbin-Watson test must be interpreted together with the autocorrelation function (ACF) plot and the partial autocorrelation function (PACF) plot.

The autocorrelation function (ACF) plot measures the correlation between an observation and its past values (lags) over time. The partial autocorrelation function (PACF) plot, on the other hand, measures the direct correlation between an observation and its past values (lags) while removing the influence of intermediate lags. The PACF plot is particularly useful for determining the appropriate order of an AR (autoregressive) model [55,56,60,63,64].

The selection of the best-fitting model considered the criteria of parsimony and model fit quality, including deviance and residual deviance [56,63]. After identifying the best-fitting model, we presented the results alongside their respective relative risk (RR) values and 95% confidence intervals (CI 95%). These values were obtained by exponentiating the parameters of the quasi-Poisson model [56,60,61,62,63].

We conducted the analyses with a significance level of *p* < 0.05 and used the lmtest, Epi, tsModel, gls, splines, vcd, nlme, ggplot2, and tidyverse libraries in R version 4.4.1 [65].

In Appendix A, we present the results of the model without serial autocorrelation terms, including the coefficient, standard error, and the Durbin–Watson test. Additionally, the final model with n serial autocorrelation terms, where applicable, is shown, including the coefficient, standard error, and the autocorrelation terms, as defined by the partial autocorrelation function.

### 2.5. Ethical Aspects

This research was conducted using freely available data from SIM/DATASUS, where no identification of individual subjects occurs, and therefore, there was no need to submit the study to a Research Ethics Committee, in accordance with Article 1 of CNS Resolution No. 510, dated 7 April 2016 [66].

## 3. Results

### 3.1. Descriptive Analysis

In Brazil, the Ministry of Health documented a total of 102,081 suicides from January 2017 to December 2023, involving individuals aged 10 to 14 up to 80 years and older. Men accounted for 78.29% of these cases (n = 79,921), while women represented 21.71% (n = 22,160). The age-standardized annual suicide rate among men was 3.71 times higher than the rate among women, with 12.33 suicides per 100,000 men compared to 3.32 suicides per 100,000 women. Mortality rates progressively increased throughout the study period across both sexes and all regions, reaching their highest levels in 2023 (Figure 2 and Figure 3).

This study observed that among men, the South and Midwest regions had the highest standardized annual average suicide rates, while the Southeast and Northeast regions had the lowest rates (Figure 1). Specifically, the male suicide rates in the South and Midwest regions were 18.95 and 14.81 suicides per 100,000 men, respectively. In contrast, the Southeast and Northeast regions recorded 10.25 and 11.40 suicides per 100,000 men, respectively.

A similar pattern appeared among women. The standardized suicide rates in the South and Midwest regions were 4.82 and 4.40 suicides per 100,000 women, respectively. In the Southeast, the rate for women surpassed that of the Northeast, with rates of 2.98 and 2.68 suicides per 100,000 women, respectively.

The evaluation of the standardized monthly average rates throughout the study period revealed that this trend persisted. The ANOVA test identified statistically significant differences at the 5% level, which Tukey’s post hoc test further confirmed. However, comparisons of standardized monthly average suicide rates among males showed exceptions, i.e., the North and Northeast regions (*p* = 0.99), the Southeast and Northeast (*p* = 0.08), and the Southeast and North (*p* = 0.34). For women, Tukey’s test did not find statistically significant differences between the Southeast and North (*p* = 0.16), the Northeast (*p* = 0.08), or Brazil (*p* = 0.99) (Table 2).

Among men, the average monthly suicide rates increased with age, ranging from 0.097 (SD = 0.039) among adolescents aged 10–14 years to 1.302 (SD = 0.139) among the elderly (Table 2). The ANOVA test revealed statistically significant differences (*p* < 0.001), which Tukey’s post hoc test validated (*p* < 0.05) for all category comparisons.

Women recorded the lowest average monthly rates in the 10–14 age group, while adolescents (15–19 years) and middle-aged women (40–59 years) showed higher rates. These rates ranged from 0.109 (SD = 0.042) to 0.314 (SD = 0.080) (Figure 3 and Table 1). The ANOVA test confirmed statistically significant differences in suicide rates by age group (*p* < 0.001), and Tukey’s post hoc test supported these findings, except for comparisons between the 15–19 and 40–59 age groups (*p* = 0.99), 15–19 and 20–39 age groups (*p* = 0.15), and 20–39 and 40–59 age groups (*p* = 0.07) (Table 2).

From 2017 to 2023, we observed higher monthly average suicide rates among White men and women (1.011 [SD = 0.0134] and 0.292 [SD = 0.037], respectively) compared to Black men and women (0.821 [SD = 0.123] and 0.195 [SD = 0.037], respectively) (Table 2). The most common methods of suicide across genders were hanging, strangulation, and suffocation. In both groups, autointoxication ranked as the second most frequent method (Table 2). These differences were statistically significant (*p* < 0.001), as confirmed by Tukey’s post hoc test, except for the comparison between firearm-related suicides and autointoxication rates among men (*p* = 0.915).

### 3.2. Bivariate Analyses

The comparison of average monthly suicide rates before and after the first COVID-19 case in Brazil revealed systematically lower values in the pre-pandemic phase for most explanatory variables. Every category showed statistically significant differences. However, the 10 to 14 age group among men (*p* = 0.144), suicides involving firearms for women (*p* = 0.458), and self-poisoning for men (*p* = 0.053) did not show statistically significant increases (Table 3).

The exploratory evaluation of the time series, using rates smoothed through the LOESS method, confirmed these results. This evaluation demonstrated a reduction in suicide rates during the early months of the pandemic (March to December 2020). Following this initial decline, the rates increased, with variations depending on region and sex (Figure 4 and Figure 5). Similar patterns emerged in suicide rates analyzed by age group, method of perpetration, and race/skin color (Appendix A).

### 3.3. An Analysis of the Impact of the Pandemic on Suicides in Brazil Using Interrupted Time Series

The pandemic influenced monthly suicide rates by age group, showing similar patterns between men and women, except in the 15 to 19 age group. Men in this group did not experience a level change (RR = 0.997; *p* = 0.886). However, women showed an abrupt increase in suicide rates (RR = 1.063; *p* = 0.003). Suicide rates progressively increased for both sexes, indicating a trend change linked to the pandemic’s effects.

Young adult men and women experienced both level and trend changes. Suicide rates abruptly decreased and then progressively increased. Elderly individuals of both sexes showed an abrupt and progressive increase in suicide rates (Table 4 and Table 5).

The pandemic did not cause an immediate impact on women’s suicide rates in Brazil, regardless of region, race/skin color, or methods used (*p* > 0.05). In contrast, men’s suicide rates significantly changed with an abrupt reduction (RR < 1; *p* < 0.05). This reduction was not statistically significant in the Northeast and Midwest regions or in cases of suicide by self-poisoning. Among Black men, the pandemic caused an abrupt increase in suicide rates (RR = 1.015; *p* = 0.006) (Table 4 and Table 5).

The long-term pandemic effects (trend change) showed consistent patterns across most variables. Suicide rates progressively increased throughout the pandemic across all regions and race/skin color categories. Men experienced a progressive increase in firearm-related suicides, while self-poisoning became prominent among women. Both sexes showed a progressive rise in suicide rates related to hanging, strangulation, and suffocation (HSS) methods (RR > 1; *p* < 0.05) (Table 4 and Table 5).

## 4. Discussion

Our findings indicate higher suicide rates among women and men residing in the South, Midwest, and North regions. The most common methods were hanging, strangulation, and suffocation (HSS). Across all variables and in both genders, there was a significant increase in monthly average suicide rates following the onset of the COVID-19 pandemic.

Using interrupted time series analysis to assess the pandemic’s effects, we identified two primary impacts: level shifts and trend changes. The level shifts reflect the immediate impact of the pandemic, while trend changes capture its long-term effects. Initially, the COVID-19 pandemic caused an abrupt reduction in monthly suicide rates, particularly among men. However, this was followed by a progressive increase as the pandemic continued.

The Southern region, heavily influenced by European colonization, exhibits cultural characteristics that may contribute to high suicide rates. Factors such as individualism, productivity pressures, and social isolation in rural areas increase emotional vulnerability. The migration of workers from the South to the Midwest during the 1970s established cultural similarities between these regions. Displacement dynamics, adaptation challenges, and economic vulnerabilities likely represent additional risk factors. In these predominantly agribusiness-oriented regions, rural workers face specific stressors. Job instability, dependence on climatic conditions, and global market pressures are common. Additionally, easy access to pesticides amplifies suicide risks by providing lethal means amid adverse psychosocial conditions [67,68,69,70,71,72].

In the North and Midwest regions, demographic composition and geographic characteristics play a significant role. These regions host a high concentration of Indigenous populations and include border municipalities in the North and Midwest regions. These areas are marked by violence, instability, and international trafficking of drugs, goods, and people. Vulnerable populations, including Indigenous people, women, and children, face heightened suicide risks. The loss of traditional territories, disruption of cultural practices, and exposure to violence severely affect mental health in these communities [2,73,74,75].

In our study, the pandemic precipitated an abrupt change, leading to a reduction in monthly suicide rates only among men across all regions. The reduction in suicide rates in Brazil was also documented in other studies conducted during the first and second years of the pandemic [2,34,39].

During wars, epidemics, pandemics, natural disasters, and terrorist attacks, social cohesion temporarily increases and may help mitigate suicide risk [11,26,27,28,76,77,78]. Min et al. [35] argues that social distancing and anxiety due to the rapid surge in confirmed cases and deaths during the early waves of the pandemic may have strengthened social cohesion and altruism, resulting in fewer instances of suicidal behavior. Furthermore, governmental measures offering financial assistance to vulnerable populations likely contributed to the reduction in suicides [22,33,35,79].

Another important factor in the reduction of suicides was the increased time spent at home with family members, given that the majority of suicides typically occur within households [32,33,35,67,68,69,70,79]. Furthermore, movement restrictions during the early waves of the pandemic contributed to a decline in suicides in public spaces [2,22,32,33,35,67,68,69,70,79]. Additionally, the excess mortality directly attributable to COVID-19 may have impacted individuals at high risk of suicide, indirectly contributing to a reduction in deaths from this health issue [79].

The differential impact of the pandemic on suicide rates among men varied significantly by race and skin color. White men exhibited a 7.2% reduction in monthly suicide rates at the onset of the pandemic, whereas Black men experienced a 1.5% increase.

The legacy of over three centuries of African slavery has profoundly shaped Brazil’s socio-cultural structures. During this historical period, societal norms assigned value to individuals based on their proximity to the White European phenotype, thereby institutionalizing anti-Black racism and systemic inequities [52,80]. Consequently, Afrodescendant populations continue to experience disproportionately adverse outcomes in key areas such as health, education, employment, and income. These pre-existing disparities were exacerbated during the economic crisis and fiscal adjustments spanning 2014 to 2019, with further intensification observed during the pandemic [52,80].

Remarkably, across all regions, race/skin color and most age groups in both sexes exhibited a noticeable shift in trends, with progressive increases in monthly suicide rates, especially from 2021 onwards. In our study, we observed the continuation of an upward trend in monthly suicide rates in the months following the official end of the COVID-19 pandemic health emergency, declared on 22 April 2022 (from May 2022 to December 2023). These results align with observations from Japan [22] and another Brazilian study that analyzed suicide rates until the second year of the pandemic [2,31,34].

Brazil did not implement a unified mental health policy to mitigate the pandemic’s effects in this area. Instead, the Psychosocial Care Network (RAPS) faced interruptions in its activities due to efforts concentrated on pandemic control. Additionally, the mental health policy in effect at the time prioritized therapeutic communities and psychiatric hospitals. This approach resulted in reduced funding for RAPS, which comprises various territorial and community-based services that are free and universally accessible [2,31,34].

The lack of adequate mental health care, combined with the pandemic’s economic and psychological impacts, contributed to a rise in the prevalence of mental disorders. Consequently, suicide rates increased. We believe this trend may persist in the post-pandemic years [11,22,23,34,35].

During the pandemic, monthly suicide rates increased progressively among both men and women across most age groups, with the exception of girls aged 10 to 14. These trends align with patterns observed in other countries during the COVID-19 pandemic, natural disasters, and public health crises. Such events often intensify stressors that elevate mental health risks [11,22,23,27,28,29,34].

In Brazil, there were distinct patterns in monthly suicide rates between men and women. Among women, adolescents aged 15 to 19 and middle-aged individuals aged 40 to 59 experienced higher rates. For men, suicide rates rose with age, peaking in the elderly population aged 60 or older.

Younger individuals often lack effective coping mechanisms to handle the stress associated with the pandemic. This contributed to an increased prevalence of mental health issues and suicide attempts [77,78].

Brazilian women faced significant hardships during this time, including income loss, greater unpaid domestic responsibilities due to school and daycare closures, and grief over the deaths of financially critical family members [2,22,33,69,70]. Prolonged proximity to domestic abusers also heightened their exposure to violence, further worsening the pandemic’s impact on mental health [81,82,83].

For young and middle-aged men, the progressive rise in monthly suicide rates during the pandemic appears linked to unemployment and income loss. These factors deteriorated living conditions for their families [75,77,78,80]. Economic and health crises are often associated with higher suicide risks and deaths of despair in this demographic. The reduction in and eventual withdrawal of financial aid policies during the second year of the pandemic likely exacerbated these economic vulnerabilities [11,22,23,27,28,29,34].

The elderly population faced unique vulnerabilities during the pandemic, marked by a sharp increase in monthly suicide rates that persisted throughout the pandemic [75,77,78]. This group, at primary risk for severe COVID-19 complications and death, likely experienced exacerbated mental distress and suicidal behavior. Brazilian elderly people’s lower educational level and limited digital literacy made it difficult for them to communicate with friends, family, and health services, intensifying feelings of isolation, loneliness, and despair, further amplifying their mental health challenges [25,31,34,39].

Suicide method choice typically reflects socio-cultural acceptability tied to normative gender roles and the accessibility of means [40,41,42]. In this study, hanging, strangulation, and suffocation (HSS) was remarkably the most frequently used method for both men and women. However, notable differences emerged in the second most common methods. Men predominantly used firearms, whereas women primarily opted for self-poisoning.

We also identified a progressive increase in suicide rates by HSS for both sexes throughout the pandemic. However, there was no detectable effect of the pandemic—neither in level nor in trend—on firearm-related suicides among women or suicides by self-poisoning among men.

Assessing the effect of the pandemic on monthly suicide rates by method is highly relevant for public health measures. This focus enables policymakers to evaluate access control strategies for suicide methods and recognize patterns of cultural acceptability associated with these methods [1,2].

Brazil successfully reduced firearm suicide risks by implementing restrictive policies on weapon and ammunition sales, as well as regulating firearm registration and possession nationally. However, under Jair Bolsonaro’s administration, these policies were relaxed, facilitating easier firearm purchases and ownership. This shift likely contributed to the rise in firearm-related suicides among men during the pandemic. Additionally, greater household firearm availability may elevate post-pandemic suicide risks [1,2,44,45,46,47,48].

The findings underscore significant public health challenges linked to the regional, demographic, and pandemic-driven variations in suicide rates across Brazil. These challenges highlight the urgent need for targeted interventions and systemic reforms to address the multifaceted nature of suicide risk factors [31,34,35].

Firstly, the pronounced geographic disparities, with higher suicide rates in the South, Midwest, and North regions, reflect deep-seated structural and cultural vulnerabilities. In regions like the North and Midwest, where Indigenous populations face systemic inequities, policies must address the compounded effects of cultural displacement, violence, and economic deprivation [2,26].

The pandemic’s impact further magnified existing vulnerabilities. While initial reductions in suicide rates may be attributed to increased social cohesion and government financial assistance, the subsequent rise underscores the long-term psychological toll of economic instability, grief, and social isolation. These findings suggest that public health strategies must extend beyond the immediate crisis to sustain mental health support during the prolonged recovery phase [2,31,34].

This study also highlights the critical influence of socio-economic factors. Racial disparities, particularly the disproportionate increase in suicide rates among Black men, point to systemic racism’s enduring impact on mental health. Policies aimed at reducing economic inequalities and ensuring equitable access to mental health care are essential to mitigate these disparities [2,26,52,80]. Similarly, the unique challenges faced by women, such as increased exposure to domestic violence and caregiving burdens during the pandemic, demand gender-sensitive interventions [31,34,35,44,81,82].

The role of method-specific suicide trends emphasizes the importance of regulating access to lethal means. The increase in firearm-related suicides during Bolsonaro’s administration reveals the detrimental effects of relaxed firearm regulations. Restrictive policies on weapon access must be reinstated and strictly enforced to prevent further escalation. The high prevalence of hanging, strangulation, and suffocation (HSS) as suicide methods highlights the need for culturally informed prevention strategies that address both access and socio-cultural acceptability [1,2,44,45,46,47,48].

Finally, the structural deficiencies in Brazil’s mental health care system, exacerbated by the pandemic, underscore the necessity of strengthening the Psychosocial Care Network (RAPS). Although recent policy changes to restore funding are promising, regional disparities in the availability of specialized services, particularly in the North and Midwest, must be addressed. Expanding the reach of primary health care to include robust mental health support and investing in community-based interventions are critical to reducing suicide risks across diverse populations.

These findings call for a multi-pronged public health response that integrates socio-economic reforms, culturally tailored interventions, and systemic improvements in mental health care delivery. By addressing the root causes and structural determinants of suicide, policymakers can mitigate the long-term mental health burden and enhance resilience in vulnerable populations.

### Strengths and Limitations of the Study

Our study presents two main limitations. The first relates to the quality of information in Brazil’s Mortality Information System regarding suicide records. To address this issue, we chose to analyze data from 2017 to 2023. This decision was influenced by significant improvements in the system’s quality over the last decade [2].

The second limitation pertains to the method employed in this study. This method did not allow for identifying the underlying causes or mechanisms linked to suicides during the pandemic [56,57,58,59,60,61,62,63]. Despite this, the interrupted time series (ITS) method enabled the analysis of the pandemic’s effect on the temporal evolution of monthly suicide rates. It also facilitated the formulation of hypotheses about factors potentially associated with changes in levels and trends.

Our findings align with the effects observed during other health crises, natural disasters, and the COVID-19 pandemic on the temporal evolution of suicides. During the modeling process, we analyzed the presence of seasonality and serial autocorrelation in the residuals, which enhances the robustness of these findings.

Moreover, regional inequalities in the availability of mental health services exacerbate the situation. While the South and Midwest regions have more developed economic infrastructures, rural and low-income populations remain underserved. In the North, disorganized urbanization and a lack of effective public policies increase marginalization and social problems, such as urban violence, which directly affect mental health.

Considering the findings of our study, it is noteworthy that men residing in Brazil, across all regions, exhibited a reduction in monthly suicide rates as an initial effect of the pandemic. However, among women, this decrease was only observed in the North and Midwest regions. Brazilian women, in general, have been more intensely impacted by the pandemic, suffering from income loss, increased unpaid domestic work due to the closure of daycare centers and schools, and the loss of family members, particularly if those individuals had a significant role in contributing to the household income [2,22,33]. Furthermore, extended proximity to domestic abusers increased their exposure to domestic violence, exacerbating the pandemic’s effects on their mental health [82,83,84,85]. In light of this, the increase in solidarity networks during this period likely had a limited effect in mitigating suicide rates among women residing in Brazil at the onset of the pandemic.

## 5. Conclusions

The findings of this study indicate that the pandemic influenced monthly suicide rates, with an abrupt reduction at the onset of the pandemic period among men. We identified that factors such as greater social cohesion, solidarity, and emergency financial assistance measures implemented during this initial period significantly impacted the reduction in suicide rates among men compared to women.

As the pandemic progressed, a decline in suicide-protective factors was observed. As a long-term effect, trend rates increased and persisted even after the pandemic ended (from May 2022 to December 2023). In this context, it is imperative for the Brazilian government to develop effective intersectoral suicide prevention strategies that consider the specificities and needs of the population in each of the country’s major regions.

It is important to intensify investments in the Psychosocial Care Network services, aiming to reduce regional disparities in access to mental health care and to expand the healthcare system’s capacity to address priority demands related to suicide. Beyond addressing regional differences, these initiatives should engage civil society and adopt a comprehensive approach that includes racial and age-related issues, as well as the most prevalent suicide methods in the country. The implementation of these strategies is essential to mitigate the pandemic’s impacts on suicide risk and to promote the mental health of the population.

## Figures and Tables

**Figure 1 ijerph-22-00138-f001:**
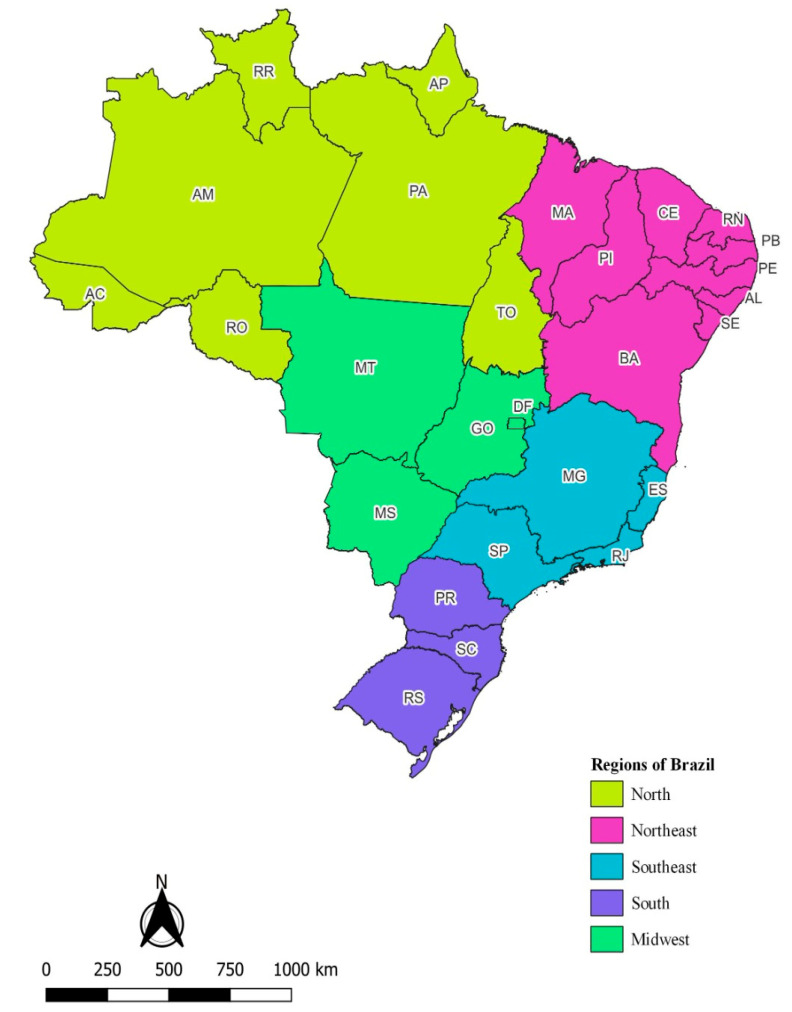
Map of Brazil, according to geographic region and states.

**Figure 2 ijerph-22-00138-f002:**
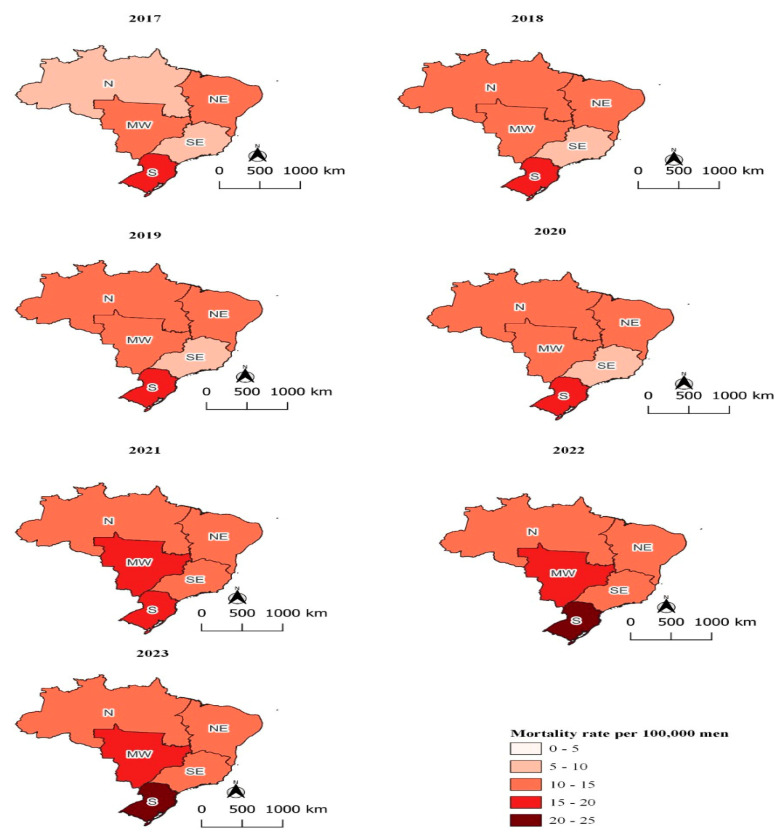
Standardized annual suicide rates per 100,000 men, by year, Brazil, 2024. Note: N—North; NE—Northeast; S—South; SE—Southeast; and MW—Midwest; source: Mortality Information System (SIM/SUS), National Bureau of Statistics (IBGE).

**Figure 3 ijerph-22-00138-f003:**
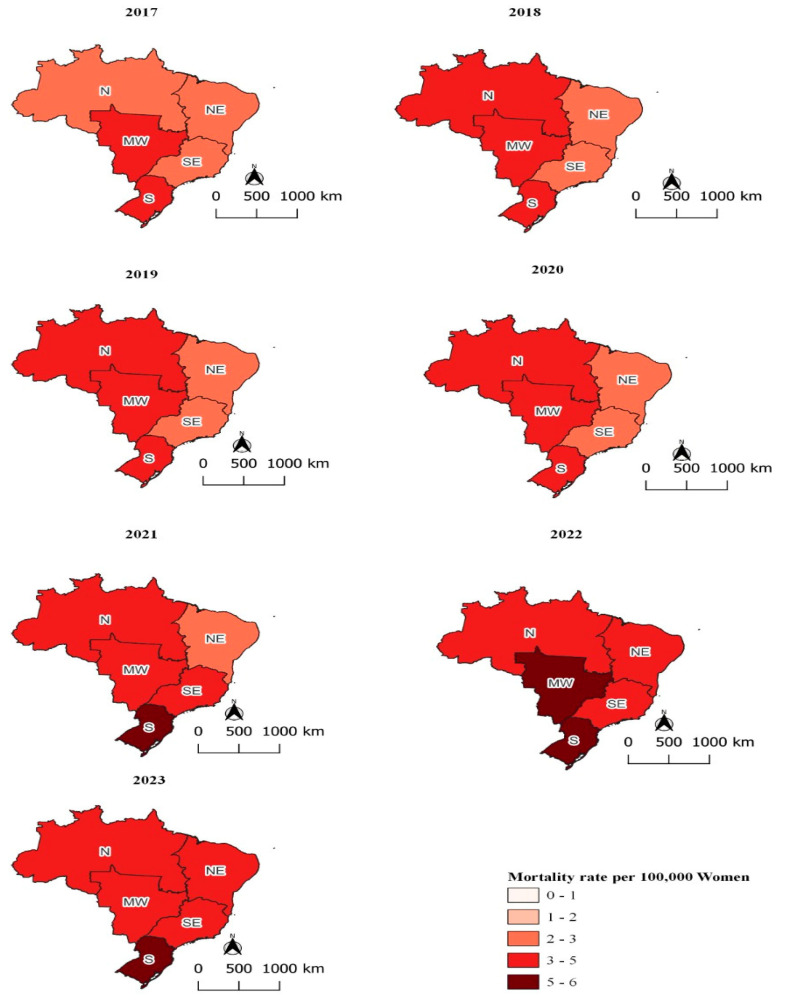
Standardized annual suicide rates per 100,000 women, by year, Brazil, 2024. Note: N—North; NE—Northeast; S—South; SE—Southeast; and MW—Midwest; source: Mortality Information System (SIM/SUS), National Bureau of Statistics (IBGE).

**Figure 4 ijerph-22-00138-f004:**
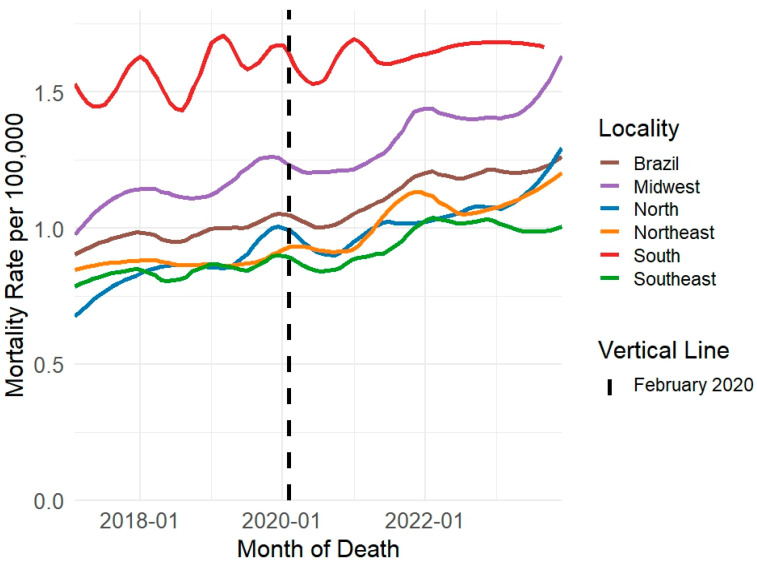
Smoothed monthly suicide rates per 100,000 men using LOESS (Locally Estimated Scatterplot Smoothing) by region (January 2017 to December 2023), Brazil, 2024. Source: Mortality Information System (SIM/SUS), National Bureau of Statistics (IBGE).

**Figure 5 ijerph-22-00138-f005:**
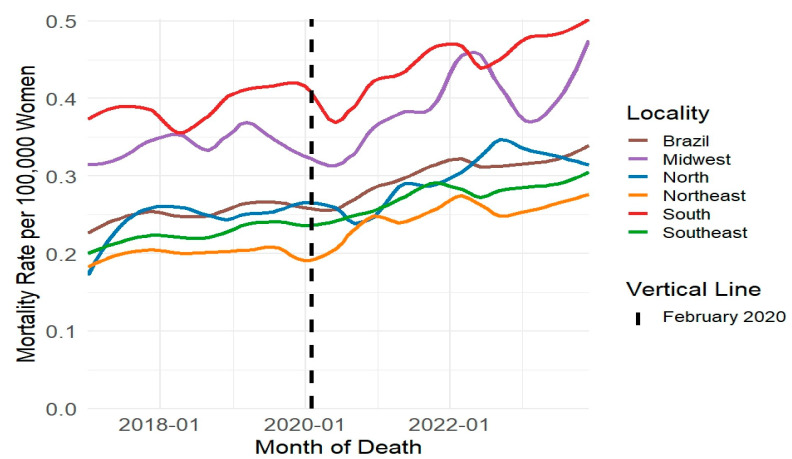
Smoothed monthly suicide rates per 100.000 women using LOESS (Locally Estimated Scatterplot Smoothing) by region (January 2017 to December 2023), Brazil, 2024. Source: Mortality Information System (SIM/SUS), National Bureau of Statistics (IBGE).

**Table 1 ijerph-22-00138-t001:** Steps in the interrupted time series modeling process for monthly suicide rates in Brazil, stratified by sex, region, age group, race/skin color, and method, from January 2017 to December 2023, Brazil 2024.

Step 1: Estimate models without seasonality and models with seasonality.
A quasi-Poisson model without seasonality (Model 1) and a quasi-Poisson model with seasonality (Model 2) are estimated. Model 1: $$\log\left(\mu_{t}\right)=\beta_{0}+\beta_{1}T+\beta_{2}X_{t}+\beta_{3}\left(T-T_{0}\right)X_{t}+offset\left(\log\left(Popt\right)\right)$$ Model 2: Seasonal effects are considered, using Fourier terms (in this case, two pairs of sine and cosine functions). $$\log\left(\mu_{t}\right)=\beta_{0}+\beta_{1}T+\beta_{2}X_{t}+\beta_{3}\left(T-T_{0}\right)X_{t}+offset\left(\log\left(Popt\right)\right)+\gamma_{1}\;sin\;\frac{2\;\pi X\left(i\right)\;}{L}$$ $$+\gamma_{2}\;cos\;\frac{2\;\pi X\left(i\right)\;}{L}+\gamma_{3}\;sin\;\frac{4\;\pi X\left(i\right)\;}{L}+\gamma_{4}\;cos\;\frac{4\;\pi X\left(i\right)\;}{L}$$ Where the Fourier terms are $\gamma_{1}\;sin\;\frac{2\;\pi X\left(i\right)\;}{L}$ $\gamma_{2}\;cos\;\frac{2\;\pi X\left(i\right)\;}{L}$ $\gamma_{3}\;sin\;\frac{4\;\pi X\left(i\right)\;}{L}$ $\gamma_{4}\;cos\;\frac{4\;\pi X\left(i\right)\;}{L}$ *L* represents the seasonality period (12 months); $\gamma_{1}e\gamma_{2}$ are coefficients associated with the fundamental frequency $\frac{2\pi X\left(i\right)}{L}$. $\gamma_{1}$ is the amplitude of the sinusoidal component of the fundamental frequency, $\gamma_{2}$ is the amplitude of the cosine component of the fundamental frequency, $\gamma_{3}e\gamma_{4}$ are coefficients associated with the first harmonic $\frac{4\pi X\left(i\right)}{L}{,\gamma }_{3}$ is the amplitude of the sinusoidal component of the first harmonic, and $\gamma_{4}$ is the amplitude of the cosine component of the first harmonic.
Step 2: Compare the models’ fit through a deviance analysis (akin to an ANOVA for GLMs) by employing an appropriate F-test.
To compare their fit, one may resort to a deviance analysis (analogous to an ANOVA for generalized linear models) using an appropriate F-test. The null hypothesis (H0) posits no significant difference in fit between the models, implying that the model without seasonality (Model 1) is adequate. If the *p*-value exceeds 0.05, H0 is not rejected, and it is concluded that the model without seasonality provides a sufficiently good fit to the data. If the *p*-value is below 0.05, H0 is rejected, and it is deduced that the model with seasonality (Model 2) achieves a superior fit.
Step 3: Assess the presence of serial autocorrelation in the residuals of the chosen model using the Durbin–Watson (DW) test as well as the autocorrelation function (ACF) and the partial autocorrelation function (PACF).
If there is no evidence of serial autocorrelation, the final model selected in the previous step is retained (Step 2). If there is evidence of serial autocorrelation, one or more autoregressive terms are added to the model selected in Step 2, as indicated by significant peaks in the PACF, resulting in an AR (autoaegressive) model. The autoregressive terms (ϕ1,ϕ2, …, ϕp) are as follows: ϕ_1_log(μt−1) includes the lag of one time unit (AR(1)). Φ_2_log(μt−2) includes the lag of two time units (AR(2)). Φplog(μt−2p) includes the lag of p time units (AR(p)).

**Table 2 ijerph-22-00138-t002:** Basic descriptive statistics on mortality rates by sex, age group, region, and race/skin color and method of perpetration. Brazil, January 2017 to December 2023, and Brazil, 2024.

Variable	Men		Women	
	Mean (SD) ^a^	*p*-Value	Mean (SD) ^a^	*p*-Value
Age group (age) ^b^				
10 to 14	0.097 (0.039)	<0.0001	0.109 (0.041)	<0.0001
15 to 19	0.714 (0.126)		0.314 (0.080)	
20 to 39	1.165 (0.175)		0.294 (0.055)	
40 to 59	1.236 (0.129)		0.317 (0.047)	
60 or more	1.310 (0.139)		0.251 (0.045)	
Locality ^b^				
North	0.958 (0.172)	<0.0001	0.246 (0.069)	<0.0001
Northeast	0.976 (0.117)		0.201 (0.029)	
Southeast	0.907 (0.109)		0.224 (0.040)	
South	1.707 (0.224)		0.394 (0.074)	
Midwest	1.267 (0.202)		0.367 (0.084)	
Brazil	1.074 (0.125)		0.281 (0.037)	
Race/skin color ^c^				
Black	0.821 (0.123)	<0.0001	0.195 (0.035)	<0.0001
White	1.011 (0.134)		0.292 (0.037)	
Method of perpetration ^b^				
Firearm	0.087 (0.012)	<0.0001	0.008 (0.002)	<0.0001
Self-poisoning	0.0919 (0.087)		0.061 (0.011)	
Hanging, strangulation, and suffocation	0.795 (0.101)		0.162 (0.023)	

^a^ Standard deviation; ^b^ ANOVA test; ^c^ Student’s test; source: Mortality Information System (SIM/SUS), National Bureau of Statistics (IBGE).

**Table 3 ijerph-22-00138-t003:** Basic descriptive statistics on mortality rates by independent variables before and during the COVID-19 pandemic. Brazil, January 2017 to December 2023, and Brazil, 2024.

Variable	Before Pandemic	After Pandemic	Men	Before Pandemic	After Pandemic	Women
	Men	Men		Women	Women	
	Mean (SD ^a^)	Mean (SD ^a^)	*p*-Value ^c^	Mean (SD ^a^)	Mean (SD ^a^)	*p*-Value ^c^
Age group (age) ^b^						
10 to 14	0.092 (0.041)	0.100 (0.037)	0.332	0.097 (0.041)	0.118 (0.037)	0.025
15 to 19	0.669 (0.138)	0.751 (0.102)	0.003	0.283 (0.066)	0.340 (0.083)	0.001
20 to 39	1.051 (0.09)	1.258 (0.172)	<0.0001	0.253 (0.035)	0.329 (0.044)	0.056
40 to 59	1.148 (0.091)	1.308 (0.110)	<0.0001	0.294 (0.030)	0.336 (0.052)	<0.0001
60 or more	1.255 (0.119)	1.355 (0.139)	0.001	0.2425 (0.048)	0.259 (0.041)	0.090
Locality ^b^						
North	0.861 (0.137)	1.039 (0.156)	<0.0001	0.246 (0.049)	0.298 (0.074)	<0.0001
Northeast	0.8762 (0.0702)	1.058 (0.117)	<0.0001	0.201 (0.042)	0.249 (0.042)	<0.0001
Southeast	0.846 (0.081)	0.956 (0.104)	<0.0001	0.224 (0.027)	0.275 (0.033)	<0.0001
South	1.583 (0.165)	1.810 (0.215)	<0.0001	0.394 (0.051)	0.445 (0.082)	<0.0001
Midwest	1.144 (0.143)	1.369 (0.188)	<0.0001	0.340 (0.06)	0.389 (0.091)	0.006
Brazil	0.9858 (0.0744)	1.147 (0.110)	<0.0001	0.253 (0.020)	0.303 (033)	<0.0001
Race/skin color ^c^						
White	0.970 (0.073)	1.046 (0.108)	0.0001	0.271 (0.028)	0.311 (0.033)	<0.0001
Black	0.722 (0.0630)	0.901 (0.097)	<0.0001	0.169 (0.021)	0.216 (0.030)	<0.0001
Methods ^b^						
Firearm	0.084 (0.011)	0.089 (0.012)	<0.0001	0.009 (0.003)	0.008 (0.002)	0.458
Self-poisoning	0.097 (0.131)	0.0877 (0.012)	0.053	0.054 (0.007)	0.065 (0.0115)	<0.0001
Hanging, strangulation, and suffocation	0.722 (0.061)	0.855 (0.0856)	<0.0001	0.147 (0.018)	0.174 (0.020)	<0.0001

^a^ Standard deviation; ^b^ ANOVA test; ^c^ Welch’s test; source: Mortality Information System (SIM/SUS) National Bureau of Statistics (IBGE).

**Table 4 ijerph-22-00138-t004:** Effect of the COVID-19 pandemic on suicide mortality rates per 100,000 men. Relative risks estimated through interrupted time series analysis and by comparing the periods January 2017 to February 2020 with March 2020 to December 2023, Brazil, 2024.

Variables	Categories	Interpretation	RR ^a^	CI95% ^b^	*p*-Value ^c^
Age group (years)	10 to 14				
	Level change	Not detected	1.142	0.805–1.620	0.459
	Trend change	Not detected	0.999	0.992–1.006	0.772
	15 to 19				
	Level change	Not detected	0.997	0.953–1.043	0.886
	Trend change	Progressive increase	1.003	1.002–1.004	<0.0001
	20 to 39				
	Level change	Abrupt reduction	0.893	0.875–0.911	<0.0001
	Trend change	Progressive increase	1.0070	1.006–1.009	<0.0001
	40 to 59				
	Level change	Not detected	1.004	0.950–1.061	0.887
	Trend change	Progressive increase	1.003	1.002–1.004	<0.0001
	60 or more years				
	Level change	Abrupt increase	1.067	1.056–1.078	<0.0001
	Trend change	Progressive increase	1.002	1.001–1.003	0.011
Locality	North				
	Level change	Abrupt reduction	0.890	0.801–0.991	0.036
	Trend change	Progressive increase	1.007	1.005–1.009	<0.0001
	Northeast				
	Level change	Not detected	1.010	0.998–1.023	0.115
	Trend change	Progressive increase	1.005	1.003–1.006	<0.0001
	Southeast				
	Level change	Abrupt reduction	0.971	0.959–0.984	<0.0001
	Trend change	Progressive increase	1.004	1.001–1.005	<0.0001
	South				
	Level change	Abrupt reduction	0.955	0.948–0.962	<0.0001
	Trend change	Progressive increase	1.004	1.003–1.005	<0.0001
	Midwest				
	Level change	Not detected	0.941	0.862–1.028	0.181
	Trend change	Progressive increase	1.006	1.004–1.007	<0.0001
	Brazil				
	Level change	Abrupt reduction	0.967	0.962–0.973	<0.0001
	Trend change	Progressive increase	1.004	1.003–1.005	<0.0001
Race/skin color	Black				
	Level change	Abrupt increase	1.015	1.005–1.026	0.006
	Trend change	Progressive increase	1.005	1.004–1.007	<0.0001
	White				
	Level change	Abrupt reduction	0.928	0.878–0.982	0.012
	Trend change	Progressive increase	1.004	1.003–1.005	<0.0001
Methods	Firearm				
	Level change	Abrupt reduction	0.894	0.806–0.992	0.037
	Trend change	Progressive increase	1.004	1.002–1.006	0.000
	Self-poisoning				
	Level change	Not detected	0.754	0.381–1.493	0.420
	Trend change	Not detected	1.005	0.991–1.019	0.529
	HSS ^d^				
	Level change	Abrupt reduction	0.981	0.973–0.989	<0.0001
	Trend change	Progressive increase	1.005	1.003–1.005	<0.0001

^a^ Relative risk; ^b^ 95% confidence interval; ^c^ interrupted time series estimated by the quasi-Poisson regression model; ^d^ hanging, strangulation, and suffocation.

**Table 5 ijerph-22-00138-t005:** Effect of the COVID-19 pandemic on suicide mortality rates per 100,000 women. Relative risks estimated through interrupted time series analysis and by comparing the periods January 2017 to February 2020 with March 2020 to December 2023, Brazil, 2024.

Variables	Categories	Interpretation	RR ^a^	CI95% ^b^	*p*-Value ^c^
Age group (years)	10 to 14				
	Level change	Not detected	1.050	0.765–1.442	0.763
	Trend change	Not detected	1.003	0.997–1.010	0.317
	15 to 19				
	Level change	Abrupt increase	1.063	1.022–1.106	0.003
	Trend change	Progressive increase	1.003	1.002–1.005	<0.0001
	20 to 39				
	Level change	Abrupt reduction	0.967	0.957–0.978	<0.0001
	Trend change	Progressive increase	1.009	1.007–1010	<0.0001
	40 to 59				
	Level change	Not detected	0.969	0.870–1.080	0.571
	Trend change	Progressive increase	1.004	1.002–1.006	0.001
	60 or more years				
	Level change	Abrupt increase	1.038	1.019–1.057	0.000
	Trend change	Progressive increase	1.002	1.0001–1.003	0.000
Locality	North				
	Level change	Not detected	0.932	0.771–1.126	0.465
	Trend change	Progressive increase	1.006	1.002–1.010	0.0021
	Northeast				
	Level change	Not detected	1.037	0.911–1.180	0.584
	Trend change	Progressive increase	1.004	1.002–1.007	0.0023
	Southeast				
	Level change	Not detected	1.051	0.964–1.145	0.267
	Trend change	Progressive increase	1.004	1.002–1.005	<0.0001
	South				
	Level change	Not detected	0.907	0.796–1.034	0.146
	Trend change	Progressive increase	1.005	1.002–1.007	0.000
	Midwest				
	Level change	Not detected	0.921	0.768–1.101	0.366
	Trend change	Progressive increase	1.005	1.002–1.009	0.007
	Brazil				
	Level change	Not detected	0.995	0.930–1.064	0.875
	Trend change	Progressive increase	1.004	1.003–1.006	<0.0001
Race/skin color	Black				
	Level change	Not detected	0.963	0.877–1.057	0.428
	Trend change	Progressive increase	1.007	1.005–1.009	<0.0001
	White				
	Level change	Not detected	1.018	0.936–1.107	0.674
	Trend change	Progressive increase	1.003	1.001–1.005	0.001
Methods	Firearm				
	Level change	Not detected	1.092	0.854–1.396	0.484
	Trend change	Not detected	0.997	0.992–1.002	0.212
	Self-poisoning				
	Level change	Not detected	1.025	0.899–1.169	0.713
	Trend change	Progressive increase	1.004	1.001–1.007	0.003
	HSS ^d^				
	Level change	Not detected	0.974	0.893–1.062	0.551
	Trend change	Progressive increase	1.005	1.003–1.006	<0.0001

^a^ Relative risk; ^b^ 95% confidence interval; ^c^ interrupted time series estimated by the quasi-Poisson regression model; ^d^ hanging, strangulation, and suffocation.

## Data Availability

https://doi.org/10.5281/zenodo.14012141.

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
