# Peer review of "Effect of the COVID-19 Pandemic on Suicide Mortality in Brazil: An Interrupted Time Series Analysis"

_ijerph, 2025, doi:10.3390/ijerph22020138_

Round 1
Reviewer 1 Report
Comments and Suggestions for Authors
Comments
General comments:
Although it was an interesting and thorough study, there were a few areas where corrections could be made to make it better.
Specific comments:
Key words:
- Please do not repeat words us used in the paper title as key words.
Abstract:
- How many cases were included in your study? Please specify this in the manuscript.
Introduction:
- While the introductory statistics on suicide rates in Brazil and other countries are valuable, they could be presented more cohesively. Consider combining details on rates with a direct comparison to clarify Brazil’s position both globally and regionally.
- Page 2, Line 43: The mention of regional disparities is important, but the sentence “for instance, the southern region recorded rates exceeding 11.0 per 100,000 in 2021 [2]” could benefit from a brief explanation of possible causes for this disparity, setting up the context for the study’s geographic analysis.
- Page 2, Lines 45-54: The paragraph listing individual and situational risk factors could be improved by categorizing these into primary themes, such as mental health, socio-economic factors, and access to lethal means, for better readability.
- Page 2, Lines 64-74: This section could use more detail about why some high-income countries saw stable or reduced suicide rates (e.g., financial support measures) while others like Brazil experienced increases. This contrast could be linked more clearly to the study’s focus on socio-economic impacts within Brazil.
- Page 2, Lines 64-74: This section could use more detail about why some high-income countries saw stable or reduced suicide rates (e.g., financial support measures) while others like Brazil experienced increases. This contrast could be linked more clearly to the study’s focus on socio-economic impacts within Brazil.
Page 3, Lines 100-101: You referenced geographic regions in your study. When addressing geographic data, it is generally expected to present findings with suitable maps to enhance comprehension. However, this technique was not applied in your study, which could be beneficial throughout, especially in the findings section.
Materials and Methods:
- My priority recommend for this study is:
"when we talk about geographic features, it is most interesting to use appropriate maps. In your case, I think using some mapping and spatial analysis techniques would improve your study results. For example, to display spatial patterns of suicide, differences between men and women's rates, total rates, spatial variations, etc.". You can search for similar study topics such as homicide or suicide rates mapping to find out the techniques that were applied. Then introduce the technologies you added and interpret them in result section (exploratory analysis or new subheading).
- Number the equations applied in your study.
Results and discussion:
- Adjust the font size used in Figure 1 to the same size, please. You also can use maps instead of charts.
In my opinion, the 'Discussion' section is too long and it is suggested to summarize it if applicable.
Thanks,
Reviewer,
Author Response
We appreciate your valuable contributions. Attached are our responses to your comments. In the revised manuscript, corrections are highlighted in red.
Specific comments:
Key words:
- Please do not repeat words us used in the paper title as key words.
Response: We changed the keywords: Suicide; Health Iniquities; Health Crises; Mental Health
Abstract:
- How many cases were included in your study? Please specify this in the manuscript.
Response: We made a suggested change.
This study analyzed the effect of the COVID-19 pandemic on suicide rates among Brazilian residents, stratified by sex. It examined monthly suicide rates using interrupted time series analysis. Researchers compared the months before the pandemic (January 2017 to February 2020) with those after the first diagnosed case of COVID-19 in Brazil (March 2020 to December 2023). They applied an interrupted time series model (quasi-Poisson) to account for serial autocorrelation in the residuals and seasonality. During this period, authorities reported 102,081 suicides in Brazil. The age-standardized annual suicide rate among men was 3.71 times higher than the rate among women (12.33 suicides per 100,000 vs. 3.32 suicides per 100,000 women). The South and Central-West regions had the highest standardized annual average suicide rates. Suicide rates among men dropped abruptly at the pandemic's onset (RR < 1, p < 0.05). However, Black men, women aged 15 to 19 years, and elderly individuals exhibited a significant increase (RR > 1, p < 0.05).Over time, suicide rates rose across most age groups, regions, and methods studied (RR > 1, p < 0.05). The pandemic’s impact differed significantly between men and women based on stratification variables. Nonetheless, a progressive upward trend emerged throughout the pandemic.
Introduction:
- While the introductory statistics on suicide rates in Brazil and other countries are valuable, they could be presented more cohesively. Consider combining details on rates with a direct comparison to clarify Brazil’s position both globally and regionally.
Response: Suggestion accepted, the paragraph was reorganized.
Suicide represents a complex and multifaceted social issue and poses a significant global public health challenge. Each year, approximately 700,000 lives are lost to suicide worldwide. Brazil ranks among the top ten countries with the highest absolute number of suicides, reporting 15,507 cases in 2021. The country's suicide rate is 7.5 deaths per 100,000 inhabitants [1]. Although this rate is lower than those observed in many Global North countries, such as those in Europe, North America, and Asia, where rates often exceed 15 per 100,000, Brazil surpasses countries in South America like Argentina (8.4/100,000), Colombia (3.9/100,000), and Peru (2.8/100,000). However, smaller nations like Guyana (40.3/100,000) and Suriname (25.4/100,000) exhibit significantly higher rates [2].
While suicide rates have declined in many regions globally, they have risen in Brazil and the Americas in recent years [1-2]. Within Brazil, regional disparities are pronounced. For example, in 2021, the South region reported 11.0 deaths per 100,000 inhabitants, whereas the Southeast region reported 6.34 per 100,000 [2]. Variations in cultural factors, lifestyle differences, and disparities in health information quality contribute to these differences.
- Page 2, Line 43: The mention of regional disparities is important, but the sentence “for instance, the southern region recorded rates exceeding 11.0 per 100,000 in 2021 [2]” could benefit from a brief explanation of possible causes for this disparity, setting up the context for the study’s geographic analysis.
Response: Suggestion accepted, a brief explanation was added.
Suicidal behavior is linked to a combination of individual and situational risk factors. At the individual level, mental disorders such as depression, bipolar disorder, schizophrenia, and substance abuse are prominent. Situational factors, including socioeconomic inequalities, limited access to healthcare, economic crises, and health emergencies, also play significant roles. Moreover, the availability of lethal means, such as firearms and toxic pesticides, increases the risk of suicide [3-10].
- Page 2, Lines 45-54: The paragraph listing individual and situational risk factors could be improved by categorizing these into primary themes, such as mental health, socio-economic factors, and access to lethal means, for better readability.
Response: Suggestion accepted, the themes were categorized and presented in a more readable way.
Suicidal behavior is associated with risk factors at both individual and situational levels, which interact to increase the risk of this health condition. Among the individual factors, mental disorders such as depression, bipolar disorder, schizophrenia, and substance abuse are prominent. At the situational level, socioeconomic factors, such as inequalities in access to employment and income, difficulties in accessing healthcare services, and economic and health crises, also play a significant role. Additionally, easy access to and availability of lethal means, such as firearms and toxic pesticides, significantly contribute to the risk of suicide [3-10].
- Page 2, Lines 64-74: This section could use more detail about why some high-income countries saw stable or reduced suicide rates (e.g., financial support measures) while others like Brazil experienced increases. This contrast could be linked more clearly to the study’s focus on socio-economic impacts within Brazil.
The pandemic’s impact on suicide rates was uneven across countries, influenced by socio-economic and cultural characteristics, as well as individual factors. High-income countries that quickly implemented financial support measures for their populations, such as Germany, Australia, Canada, England, South Korea, Japan, Norway, and Taiwan, often observed reductions or stabilization in suicide rates [19-23]. Investments in social welfare, healthcare, and employment programs in these nations helped mitigate the mental health impacts of economic crises. For example, in Europe and USA, each additional $10 per capita invested in employment programs during economic downturns reduced unemployment’s impact on suicide rates [19-23].
Conversely, in countries such as Nepal and Brazil, suicide rates increased during the initial months of the pandemic [24–25]. Unemployment and poverty, which tend to worsen during such periods, appear to have significantly contributed to this rise. Evidence also shows that after public health crises, natural disasters, terrorist attacks, and wars, suicide rates often increase over time. This pattern is particularly evident in low- and middle-income countries, where public support policies are frequently delayed in their implementation [11,22,26–29].
In Brazil, the government's income redistribution policy (emergency aid) was implemented late. Moreover, accessing this financial assistance required individuals to register through a mobile phone application. This requirement posed a significant barrier for populations lacking internet access. Compounding these challenges, during the second year of the pandemic, the government reduced the value of this financial aid and eventually discontinued it altogether. These factors likely exacerbated economic hardships and contributed to the observed rise in suicide rates.
Page 3, Lines 100-101: You referenced geographic regions in your study. When addressing geographic data, it is generally expected to present findings with suitable maps to enhance comprehension. However, this technique was not applied in your study, which could be beneficial throughout, especially in the findings section.
Materials and Methods:
- My priority recommend for this study is:
"when we talk about geographic features, it is most interesting to use appropriate maps. In your case, I think using some mapping and spatial analysis techniques would improve your study results. For example, to display spatial patterns of suicide, differences between men and women's rates, total rates, spatial variations, etc.". You can search for similar study topics such as homicide or suicide rates mapping to find out the techniques that were applied. Then introduce the technologies you added and interpret them in result section (exploratory analysis or new subheading).
Response: We appreciate the suggestions provided. A map displaying the division of Brazil by Federative Unit and region has been included. The results initially presented in Figure 1 have been divided into two separate figures (Figure 2 e Figure 3) to show the annual rates (2017 to 2023) by sex.
The smoothed rates, calculated using the LOESS method, have been maintained as line graphs due to the dataset spanning 84 months. This approach allows for a more appropriate presentation, enabling exploratory evaluation of trends in monthly suicide rates before and after the COVID-19 pandemic by region, age group, method, and race/skin color.
It is important to highlight that, to achieve the objective of our study— “This study analyzed the effect of the COVID-19 pandemic on suicide rates among Brazilian residents, stratified by sex”—the interrupted time series method is the most suitable analytical approach. This method facilitates the analysis of monthly rate trends before and during the intervention (the COVID-19 pandemic). Consequently, spatial analysis techniques, such as Local and Global Moran’s Index, do not align with the study's objectives. Furthermore, the analysis does not focus on rates by Federative Unit but rather on geographic regions (n=5), rendering spatial analyses unfeasible.
- Number the equations applied in your study.
Response: We appreciate the suggestion and have incorporated it into the manuscript. Specifically, we have included the equation for the interrupted time series model with the Fourier terms. Additionally, we have added a table detailing the steps of the modeling process, which includes the relevant autoregressive terms.
Results and discussion:
a)Adjust the font size used in Figure 1 to the same size, please. You also can use maps instead of charts.
Response: We appreciate the comment and have modified the figure, splitting it into two. In these figures, the annual rates are presented on a map with the regional divisions of Brazil. Each map in the figure represents the rate for a specific year.
b)In my opinion, the 'Discussion' section is too long and it is suggested to summarize it if applicable.
Response: We appreciate the suggestion. During the English revision, we made an effort to enhance the text's cohesion and removed redundant information. However, following the recommendations of Reviewer 2, it was necessary to add several paragraphs to include a discussion on the role of mental health services.
Reviewer 2 Report
Comments and Suggestions for Authors
1. Provide more detailed explanations of the statistical methods used
2. Exploring socio-economic, cultural, and healthcare access differences could add depth to the analysis.
3. Consider extending the analysis to include post-pandemic trends beyond December 2022
4. Highlight the importance of mental health services during and after the pandemic.
5. Discuss the broader public health implications of the findings
Comments on the Quality of English Language1. Some sentences are quite long and complex. Breaking them into shorter, more straightforward sentences can improve comprehension.
2. Use the active voice to make the writing more direct and engaging.
3. The English could be improved to more clearly express the research.
Author Response
We appreciate your valuable contributions. We have conducted an extensive revision of the English text, following your suggestions. As shown in the revised file, we have entirely removed the previous text and replaced it with the revised version, highlighting the modifications in red.
Revisor 2
Comments and Suggestions for Authors
We appreciate the suggestions, which will undoubtedly contribute to the improvement of our manuscript. The corrections made are highlighted in red.
- Provide more detailed explanations of the statistical methods used.
Response:We appreciate the suggestion. In the methodology section, we have expanded the description of the statistical tests used in the bivariate analysis. Additionally, we have provided a more detailed explanation of the smoothing of rates using the LOESS method. We have elaborated on the STI method applied, incorporating the equation of the model that accounts for seasonality through Fourier terms. Furthermore, we have detailed the tests employed to assess the presence of serial autocorrelation, specifically the Durbin-Watson test, as well as the examination of residuals using the autocorrelation function (ACF) and the partial autocorrelation function (PACF) plots.To facilitate the reader’s understanding of the modeling process, we have included Table 1, titled "Steps in the Interrupted Time Series Modeling Process":Step 1: Estimate models without seasonality and models with seasonality;Step 2: Compare the fit of these models through a deviance analysis (analogous to ANOVA for generalized linear models), using an appropriate F-test; and Step 3: Assess the presence of serial autocorrelation in the residuals of the chosen model using the Durbin-Watson (DW) test, along with the autocorrelation function (ACF) and the partial autocorrelation function (PACF).
In Supplementary Tables S1 and S2, we presented the results of the model without serial autocorrelation terms, including the Coefficient, Standard Error, and the Durbin-Watson test. Additionally, the Final model with n serial autocorrelation terms, where applicable, is shown, including the Coefficient, Standard Error, and the Autocorrelation terms as defined by the partial autocorrelation function
- Exploring socio-economic, cultural, and healthcare access differences could add depth to the analysis.
We reorganized the part of the article that already addressed socioeconomic and cultural issues to meet the reviewer's suggestion.
The Southern region, heavily influenced by European colonization, exhibits cultural characteristics that may contribute to high suicide rates. Factors such as individualism, productivity pressures, and social isolation in rural areas increase emotional vulnerability. The migration of workers from the South to the Midwest during the 1970s established cultural similarities between these regions. Displacement dynamics, adaptation challenges, and economic vulnerabilities likely represent additional risk factors. In these predominantly agribusiness-oriented regions, rural workers face specific stressors. Job instability, dependence on climatic conditions, and global market pressures are common. Additionally, easy access to pesticides amplifies suicide risks by providing lethal means amid adverse psychosocial conditions [71-72].
In the Northern and Midwest regions, demographic composition and geographic characteristics play a significant role. These regions host a high concentration of Indigenous populations and include border municipalities in the Northern and Central Arcs. These areas are marked by violence, instability, and international trafficking of drugs, goods, and people. Vulnerable populations, including Indigenous people, women, and children, face heightened suicide risks. The loss of traditional territories, disruption of cultural practices, and exposure to violence severely affect mental health in these communities [2,73-75].
In our study, the pandemic precipitated an abrupt change, leading to a reduction in monthly suicide rates only among men across all regions. The reduction in suicide rates in Brazil was also documented in other studies conducted during the first and second years of the pandemic [2,34,39].
During wars, epidemics, pandemics, natural disasters, and terrorist attacks, social cohesion temporarily increases and may help mitigate suicide risk [11,26-28,76-78]. Min et al. [35] argues that social distancing and anxiety due to the rapid surge in confirmed cases and deaths during the early waves of the pandemic may have strengthened social cohesion and altruism, resulting in fewer instances of suicidal behavior. Furthermore, governmental measures offering financial assistance to vulnerable populations likely contributed to the reduction in suicides [22,33,35,79].
Another important factor in the reduction of suicides was the increased time spent at home with family members, given that the majority of suicides typically occur within households [32-33, 35, 67-70, 79]. Furthermore, movement restrictions during the early waves of the pandemic contributed to a decline in suicides in public spaces [2, 22, 32-33, 35, 67-70, 79]. Additionally, the excess mortality directly attributable to COVID-19 may have impacted individuals at high risk of suicide, indirectly contributing to a reduction in deaths from this health issue [79].
- Consider extending the analysis to include post-pandemic trends beyond December 2022
Response: We appreciate the suggestion and have included the months from January to December 2023. We would like to highlight that, in Brazil's mortality information system, there are still no death records available for the year 2024. The inclusion of the months from 2023 confirmed the progressive increase in monthly suicide rates in the period following the pandemic. These findings align with those of other studies, which demonstrate a rise in suicide rates in the years following health crises, terrorist attacks, wars, and major disasters.
- Highlight the importance of mental health services during and after the pandemic.
Response: We have included those paragraphs in the article to address the reviewer's suggestion.
Brazil did not implement a unified mental health policy to mitigate the pandemic's effects in this area. Instead, the Psychosocial Care Network (RAPS) faced interruptions in its activities due to efforts concentrated on pandemic control. Additionally, the mental health policy in effect at the time prioritized therapeutic communities and psychiatric hospitals. This approach resulted in reduced funding for RAPS, which comprises various territorial and community-based services that are free and universally accessible [2,31,34].The lack of adequate mental health care, combined with the pandemic's economic and psychological impacts, contributed to a rise in the prevalence of mental disorders. Consequently, suicide rates increased. We believe this trend may persist in the post-pandemic years [11,22-23,34-35].
- Discuss the broader public health implications of the findings
Response: We have included those paragraphs in the article to address the reviewer's suggestion.
The findings underscore significant public health challenges linked to the regional, demographic, and pandemic-driven variations in suicide rates across Brazil. These challenges highlight the urgent need for targeted interventions and systemic reforms to address the multifaceted nature of suicide risk factors [31,34-35].
Firstly, the pronounced geographic disparities, with higher suicide rates in the South, Midwest, and North regions, reflect deep-seated structural and cultural vulnerabilities. In regions like the North and Midwest, where Indigenous populations face systemic inequities, policies must address the compounded effects of cultural displacement, violence, and economic deprivation [2,26].
The pandemic’s impact further magnifies existing vulnerabilities. While initial reductions in suicide rates may be attributed to increased social cohesion and government financial assistance, the subsequent rise underscores the long-term psychological toll of economic instability, grief, and social isolation. These findings suggest that public health strategies must extend beyond the immediate crisis to sustain mental health support during the prolonged recovery phase [2,31,34].
The study also highlights the critical influence of socioeconomic factors. Racial disparities, particularly the disproportionate increase in suicide rates among Black men, point to systemic racism’s enduring impact on mental health. Policies aimed at reducing economic inequalities and ensuring equitable access to mental health care are essential to mitigate these disparities [2,26,52,80]. Similarly, the unique challenges faced by women, such as increased exposure to domestic violence and caregiving burdens during the pandemic, demand gender-sensitive interventions [31,34-35,44,81-82].
The role of method-specific suicide trends emphasizes the importance of regulating access to lethal means. The increase in firearm-related suicides during Bolsonaro’s administration reveals the detrimental effects of relaxed firearm regulations. Restrictive policies on weapon access must be reinstated and strictly enforced to prevent further escalation. The high prevalence of hanging, strangulation, and suffocation (HSS) as suicide methods highlights the need for culturally informed prevention strategies that address both access and socio-cultural acceptability [1,2,44-48].
Finally, the structural deficiencies in Brazil’s mental health care system, exacerbated by the pandemic, underscore the necessity of strengthening the Psychosocial Care Network (RAPS). Although recent policy changes to restore funding are promising, regional disparities in the availability of specialized services, particularly in the North and Central-West, must be addressed. Expanding the reach of primary health care to include robust mental health support and investing in community-based interventions are critical to reducing suicide risks across diverse populations.
These findings call for a multi-pronged public health response that integrates socio-economic reforms, culturally tailored interventions, and systemic improvements in mental health care delivery. By addressing the root causes and structural determinants of suicide, policymakers can mitigate the long-term mental health burden and enhance resilience in vulnerable populations.
Response:We have included those paragraphs in the article to address the reviewer's suggestion.
6.Comments on the Quality of English Language
Some sentences are quite long and complex. Breaking them into shorter, more straightforward sentences can improve comprehension. Use the active voice to make the writing more direct and engaging. The English could be improved to more clearly express the research.
Response: We appreciate the reviewer’s detailed suggestion. We have conducted a thorough revision of the entire text, following the reviewer’s recommendations. As shown in the revised file, we have removed the previous text entirely and replaced it with the revised version, with the modifications highlighted in red.
Reviewer 3 Report
Comments and Suggestions for Authors
Congrats on the manuscript.
Improve readability on the abstract, minimising the use of comas and the word «we». It means that translation to English should be improved in the abstract session.
In line 102 is mentioned «research questions», but only read one. I don't understand the option of making the variable «gender» the (in)dependent variable. Somewhere in the text, this preference should be mentioned based on the convenience according to results of the regression model used.
I appreciate the statistic presentation (and methodology used), although, in the end, it turns out to be a descriptive study (but useful for future studies on «suicide in Brazil»).
Author Response
We appreciate your valuable contributions. Attached are our responses to your comments. In the revised manuscript, corrections are highlighted in red.
Revisor 3
Comments and Suggestions for Authors
Congrats on the manuscript.
1.Improve readability on the abstract, minimising the use of comas and the word «we». It means that translation to English should be improved in the abstract session.
Response: Thank you for your requested correction. We have made the adjustments and revised the English text. We emphasize that all corrections have been marked in red letters.
This study analyzed the effect of the COVID-19 pandemic on suicide rates among Brazilian residents, stratified by sex. It examined monthly suicide rates using interrupted time series analysis. Researchers compared the months before the pandemic (January 2017 to February 2020) with those after the first diagnosed case of COVID-19 in Brazil (March 2020 to December 2023). They applied an interrupted time series model (quasi-Poisson) to account for serial autocorrelation in the residuals and seasonality. During this period, authorities reported 102,081 suicides in Brazil. The age-standardized annual suicide rate among men was 3.71 times higher than the rate among women (12.33 suicides per 100,000 vs. 3.32 suicides per 100,000 women). The South and Central-West regions had the highest standardized annual average suicide rates. Suicide rates among men dropped abruptly at the pandemic's onset (RR < 1, p < 0.05). However, Black men, women aged 15 to 19 years, and elderly individuals exhibited a significant increase (RR > 1, p < 0.05).Over time, suicide rates rose across most age groups, regions, and methods studied (RR > 1, p < 0.05). The pandemic’s impact differed significantly between men and women based on stratification variables. Nonetheless, a progressive upward trend emerged throughout the pandemic.
2.In line 102 is mentioned «research questions», but only read one. I don't understand the option of making the variable «gender» the (in)dependent variable. Somewhere in the text, this preference should be mentioned based on the convenience according to results of the regression model used.
Response: We conducted the corrections and clarified that we have only one research question. We opted for stratified analyses by sex because men and women tend to exhibit self-destructive behaviors aligned with societal gender roles. These behaviors are influenced by hegemonic masculinities and femininities, contributing to the so-called suicide or gender paradox. In the introduction, we included a paragraph explaining these differences in suicidal behavior between men and women.
Additionally, the interrupted time series model, adjusted through segmented regression, included the following variables: the response variable was the number of suicides, and the independent variable accounted for all months from 2017 to 2023 (spanning 1 to 84 months). To evaluate level changes, we coded the pre-pandemic period as 0 and the post-pandemic months as 1 [56,62-63].
Since our goal was to analyze the pandemic's effect in Brazil across different sexes, age groups, regions, methods, and racial/ethnic categories, we estimated 16 STI models for men and 16 STI models for women, following the steps outlined in Table 1, which was included in the methodology.
In the results and discussion, we presented the similarities and differences in the pandemic's effects, highlighting level changes and trend changes between men and women across the evaluated variables (region, age group, racial/ethnic category, and methods).
3.I appreciate the statistic presentation (and methodology used), although, in the end, it turns out to be a descriptive study (but useful for future studies on «suicide in Brazil»).
Response : We appreciate the comments and suggestions, which can enhance the quality of our manuscript. At the end of the discussion, we delved deeper into how our findings could guide suicide prevention policies in Brazil. Additionally, we highlighted the strengths and limitations of our study.